# Pretrained Hybrids with MAD Skills

**Nicholas Roberts   Samuel Guo   Zhiqi Gao   Satya Sai Srinath Namburi GNVV**
**Sonia Cromp   Chengjun Wu   Chengyu Duan   Frederic Sala**
University of Wisconsin-Madison
nick11roberts@cs.wisc.edu

## Abstract

While Transformers underpin modern large language models (LMs), there is a growing list of alternative architectures with new capabilities, promises, and tradeoffs. This makes choosing the right LM architecture challenging. Recently proposed *hybrid architectures* seek a best-of-all-worlds approach that reaps the benefits of all architectures. Hybrid design is difficult for two reasons: it requires manual expert-driven search, and new hybrids must be trained from scratch. We propose **Manticore**,[1] a framework that addresses these challenges by *automating the design of hybrid architectures* while reusing pretrained models to create *pretrained* hybrids. Our approach augments ideas from differentiable Neural Architecture Search (NAS) by incorporating simple projectors that translate features between pretrained blocks from different architectures. We then fine-tune hybrids that combine pretrained models from different architecture families—such as the GPT series and Mamba—end-to-end. With Manticore, we enable LM selection without training multiple models, the construction of pretrained hybrids from existing pretrained models, and the ability to *program* pretrained hybrids to have certain capabilities. Manticore hybrids match existing manually designed hybrids, achieve strong performance on the Long Range Arena benchmark, and improve on pretrained transformers and state space models on various natural language tasks.

## 1 Introduction

Transformers are the workhorse architecture for large language models and beyond, powering a vast collection of foundation models. While for years it appeared that the Transformers family would remain the undisputed standard, a recent *Cambrian explosion* of proposed architectures has taken place. Many of the new architectures achieve subquadratic complexity—in contrast to the quadratic complexity of self-attention in Transformers—by using local or linear attention (De et al., 2024; Botev et al., 2024; Arora et al., 2024; Zhang et al., 2024), resurrecting and scaling recurrent networks (Botev et al., 2024; De et al., 2024; Peng et al., 2023), or by building on state-space modeling principles (Gu & Dao, 2023; Poli et al., 2023b;a; Fu et al., 2023; Gu et al., 2022). These approaches potentially promise to overturn the dominance of Transformers through more efficient training and inference.

However, no single new model is a clear overall winner when varying data modalities, tasks, and model sizes. Comparing architectures on a fixed task is fraught with difficulties (Amos et al., 2024). Even if these are overcome, practitioners would have to experiment with and evaluate every architecture for each new task—an expensive proposition. Instead, seeking a best-of-all-worlds approach, researchers have proposed the use of *hybrid models* that mix multiple architectures. These hybrids, such as the MambaFormer (Park et al., 2024)—a mix of the popular SSM Mamba architecture with a standard Transformer—have shown potential in maintaining the desirable properties of multiple model classes.

While promising, hybrids suffer from two main obstacles that stymie their adoption:

---

[1]The Manticore is a fearsome human/lion/scorpion hybrid from Persian mythology.

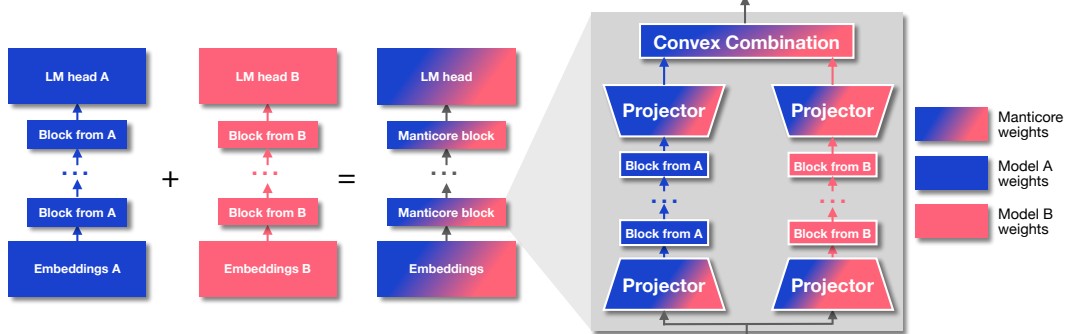

Figure 1: Manticore enables: (1) cross-architecture LM selection, (2) the construction of pretrained hybrids, and (3) the ability to program hybrids to have certain skills.

- **Manual Design.** Hybrid architectures are hand-crafted, either by manually exploring the large search space of hybrids or by relying on often unreliable intuition and heuristics.
- **Failure to Use Pretrained Models.** It is unclear how to integrate *pretrained* components from models with different architectures. Pretrained models are a key advantage of foundation models, but due to compatibility issues, hybrids are often trained from scratch, which is both limiting and costly.

A potential solution to the latter challenge is the use of *model merging* (Yadav et al., 2023; Yu et al., 2023; Wortsman et al., 2022; Ilharco et al., 2023; Davari & Belilovsky, 2023; Jang et al., 2024), some of which can operate cross-architecture (Akiba et al., 2024; Goddard et al., 2024). Unfortunately, such tools are embryonic–they are expensive and it is unclear how well they work with the diverse architectures a user may seek to build a hybrid from.

We propose a framework for automatically designing hybrid architectures that overcomes these obstacles. Our approach is inspired by principles from neural architecture search (NAS), but applies these at the level of *LM blocks* rather than convolutional cells (Liu et al., 2019; Li et al., 2021) or operations (Shen et al., 2022; Roberts et al., 2021). The resulting framework is simple, tractable, and it sidesteps merging different architectures by using simple projectors to translate between the "languages" spoken by various architectures. This enables us to include blocks from many different architectures/models with no changes required. Furthermore, inspired by the mechanistic architecture design framework (MAD) (Poli et al., 2024), we show how to learn hybrids via MAD that transfer to new tasks.

Concretely, with our proposed system, Manticore, we:

1. **Automatically select** language models, without training several models from scratch,
2. **Automatically construct** pretrained hybrids without evaluating the entire search space,
3. Explore when it is possible to **program hybrids** without full training.

Experimentally, our automatically designed hybrids compete with existing hybrids and models on the MAD tasks (Poli et al., 2024) and Long Range Arena (Tay et al., 2021, LRA), we produce pretrained hybrids that improve downstream fine-tuning performance on a variety of language tasks, and we show that Manticore can be programmed using MAD.

## 2 Methods

We now describe Manticore, our framework for automatically designing hybrid architectures by mixing components of pretrained models. We rely on projectors to align features across architectures, then apply a convex combination to aligned features, as shown in Figure 1.

In Section 2.1, we discuss and formally define the structure of Manticore hybrids: the projectors and convex combination mixture weights, as well as how both of these components are used within Manticore. In Section 2.2, we detail the NAS-inspired search procedures and

training routines involved in pretraining, fine-tuning, and programming hybrids. Finally, we provide the synthetic and real data settings that we use in our experiments in Section 3.

### 2.1 The Structure of Manticore Hybrids

Our framework comprises three main parts: the individual LMs that we combine to produce our overall hybrid, projectors that translate feature representations between LMs of different architectures, and convex combination mixture weights that specify how much the hybrid will use the features of each component architecture. We detail each of these in the following.

**Component Models** We refer to a model that is used in Manticore as a *component model*. Any modern decoder-only LM can be used as a component model in our framework. In this section, we will formally define the general high-level structure of the component models that we support. For an LM $M$ with model embedding dimension $d_M$ on a sequence of $t$ tokens from a set $\mathcal{V}$, denoted $x = (x_1, ..., x_t) \in \mathcal{V}^t$, a forward pass $M(x)$ is typically computed using the following recipe:

1. Apply an embedding function, $M_{\text{embed}} : \mathcal{V}^t \to \mathbb{R}^{t \times d_M}$ to the tokens, resulting in a sequence of embeddings denoted $x_{\text{embed}} = M_{\text{embed}}(x)$.

2. Take forward passes through $L_M$ 'blocks'–we denote the $\ell^{\text{th}}$ block as $M_{\text{Block}}^{(\ell)} : \mathbb{R}^{t \times d_M} \to \mathbb{R}^{t \times d_M}$. Specifically, for all $\ell \in [L_M]$, we obtain $x_{\ell+1} = M_{\text{Block}}^{(\ell)}(x_\ell)$, where $x_1 := x_{\text{embed}}$.

3. Finally, we pass $x_{L_M+1}$ into a language modeling head, $M_{\text{head}} : \mathbb{R}^{t \times d_M} \to (\Delta^{|\mathcal{V}|-1})^t$, where $\Delta^{|\mathcal{V}|-1}$ is the probability simplex of dimension $|\mathcal{V}|$.

This recipe applies to virtually all transformer-based LMs, recurrent models, and state-space models. Manticore supports all of these and any architecture that follows this recipe.

**Projectors** Suppose we have pretrained component models $M$ and $M'$. Assuming that the model dimensions are the same for both models ($d_M = d_{M'}$), blocks from $M$ and $M'$ may not be compatible, as their input and output features are distributed differently. It is also possible that $d_M \neq d_{M'}$, in which case composing blocks from $M$ and $M'$ is not well defined.

To overcome this issue, we apply projectors to both the inputs and the outputs of a block (or a sequence of blocks, discussed in Section 2.1) that we wish to combine in Manticore hybrids. Overall, our goal in designing projectors is to enable the blocks of $M$ and $M'$ to *share a common representation*, such that their features are compatible and can be reused in the resulting hybrid model. This is conceivably challenging—the mapping between feature spaces could be highly nonlinear and might require a lot of task-specific data to adequately learn the mapping. If the mapping is indeed highly nonlinear, we might need heavyweight multi-layer projectors with a large number of parameters. This could substantially increase parameter counts, inference cost, and could increase the data requirement for learning them. So do projectors need to be heavyweight, data-hungry, highly nonlinear objects? Fortunately, we find that the answer is no—we find that a simple linear transformation with a gated residual, pretrained on general language data, is sufficient.[2]

Suppose that we want to create a Manticore hybrid from $K$ different pretrained component models, denoted $M_{(1)}, ..., M_{(K)}$ with model dimensions $d_{M_{(1)}}, ..., d_{M_{(K)}}$. We define $d_{\max} := \max_{k \in [K]} d_{M_{(k)}}$, then want *input* and *output* projectors for the blocks of each model that convert their features to a common feature space of dimension $d_{\max}$. For any sequence of blocks of length $(n+1) < L_{d_{M_{(k)}}}$ from model $M_{(k)}$ and length-$t$ input,

$$\left( M_{(k)\text{Block}}^{(\ell+n)} \circ ... \circ M_{(k)\text{Block}}^{(\ell)} \right) : \mathbb{R}^{t \times d_{M_{(k)}}} \to \mathbb{R}^{t \times d_{M_{(k)}}},$$

---

[2]When this gating is combined with Equation 1, we see that the use of gated residuals ensures that the component architectures are still in our search space. This is a convenient property that allows Manticore to fall back on a component model when it outperforms hybrids.

we want $\text{Proj-in}_{(k)}^{(\ell)} : \mathbb{R}^{t \times d_{\max}} \to \mathbb{R}^{t \times d_{M_{(k)}}}$ and $\text{Proj-out}_{(k)}^{(\ell+n)} : \mathbb{R}^{t \times d_{M_{(k)}}} \to \mathbb{R}^{t \times d_{\max}}$, so that

$$\left( \text{Proj-out}_{(k)}^{(\ell+n)} \circ M_{(k)\text{Block}}^{(\ell+n)} \circ ... \circ M_{(k)\text{Block}}^{(\ell)} \circ \text{Proj-in}_{(k)}^{(\ell)} \right) : \mathbb{R}^{t \times d_{\max}} \to \mathbb{R}^{t \times d_{\max}}.$$

For input $x \in \mathbb{R}^{t \times d_{M_{(k)}}}$ we parameterize projectors as linear layers with gated residuals:

$$\text{Proj-in}_{(k)}^{(\ell)}(x; \alpha) := (1 - \alpha) \cdot \text{Linear}_{d_{\max} \to d_{M_{(k)}}}(x) + \alpha \cdot \text{Trunc}(x; d_{M_{(k)}})$$

$$\text{Proj-out}_{(k)}^{(\ell)}(x; \alpha) := (1 - \alpha) \cdot \text{Linear}_{d_{M_{(k)}} \to d_{\max}}(x) + \alpha \cdot \text{Pad}(x; d_{\max}).$$

Respectively, $\text{Trunc}(\cdot; d)$ and $\text{Pad}(\cdot; d)$ truncate and zero-pad input to dimension $d$, and $\text{Linear}_{d \to d'} : \mathbb{R}^d \to \mathbb{R}^{d'}$ is a learnable linear layer with gating weights $\alpha \in [0, 1]$. In total, where $\alpha \in \Delta^{K-1}$ and $I_k$ is a length-$n_k$ vector of block indices from component model $k$, we define the output of the block sequence defined by $I_k$ as

$$h_k(x; \alpha_k, I_k) = \left( \text{Proj-out}_{(k)}^{(I_{k,n_k})} \circ M_{(k)\text{Block}}^{(I_{k,n_k})} \circ ... \circ M_{(k)\text{Block}}^{(I_{k,1})} \circ \text{Proj-in}_{(k)}^{(I_{k,1})} \right)(x; \alpha_k).$$

**Mixture Weights** Next, we would like to mix the activations of different component models' block sequences, in a way that allows us to learn how much influence the blocks from each component model will have on the overall hybrid model. Learning the amount of influence that each block sequence should have on the overall hybrid is critical—if certain blocks produce less helpful features, we need a way to down-weight them. Conversely, we want to use the best blocks in our hybrid as much as possible—we want to up-weight helpful blocks. Overall, a parameterization that allows us to learn these weights should lead to better hybrids. We do this by taking a convex combination of the projectors' outputs: given the projected features $h_k(x; \alpha_k, I_k)$ for each component model $k \in [K]$, we output a convex combination of projected features

$$\text{Mix}_\alpha(x; I_1, ..., I_K) = \sum_{k \in [K]} \alpha_k h_k(x; \alpha_k, I_k). \tag{1}$$

We reuse the convex combination weights as the gating weights in the projectors. This choice yields the convenient property that when the mixture weights $\alpha$ are set to one in index $k$ and zero everywhere else, the Mix function exactly computes a sequence of blocks from component model $k$ while completely ignoring the projectors and the blocks from other component models. We adopt a popular parameterization for mixture weights from the NAS literature (Liu et al., 2019): we parameterize $\alpha$ as a softmax of a parameter vector—that is, $\alpha_k := \frac{\exp(a_k)}{\sum_{j \in [K]} \exp(a_j)}$ for all $k \in [K]$.

**Manticore** We are now ready to define our overall hybrid architecture. We seek to create a hybrid from $K$ component models, $M_{(1)}, ..., M_{(K)}$, each with a potentially different number of blocks, denoted $L_{M_{(k)}}$ for component model $k$. We fix $L$ to be the number of *Manticore blocks*, where $L$ is a common factor of each of the depths $L_{M_{(k)}}$, for all $k \in [K]$—we treat this choice of factor as a hyperparameter. For each of the $L$ Manticore blocks, we want to mix a sequence of blocks from each of the $K$ component models. We also want the number of blocks from each model $k \in [K]$ that are allocated to a single Manticore block to be evenly spread throughout the $L$ Manticore blocks—this is why we require $L$ to be a factor of $L_{M_{(k)}}$. For each component model $k \in [K]$, divide the indices of the blocks $[L_{M_{(k)}}]$ evenly into $L$ contiguous parts, denoted as $[L_{M_{(k)}}] = (I_{k,1}, ..., I_{k,L})$. Then, adopting the notation from our component models, a Manticore block is defined as

$$\text{Manticore}_{\text{Block}}^{(\ell)}(\cdot) := \text{Mix}_{\alpha^{(\ell)}}(\cdot; I_{1,\ell}, ..., I_{K,\ell})$$

with $\text{Manticore}_{\text{Block}}^{(\ell)} : \mathbb{R}^{t \times d_{\max}} \to \mathbb{R}^{t \times d_{\max}}$, for each $\ell \in [L]$, and $\alpha^{(\ell)}$ being the mixture weights at $\ell$. Next, we initialize a new set of embedding weights and a new

task specific (or language modeling) head, and we can finally illustrate a forward pass with a Manticore hybrid model, denoted using the shorthand notation $\text{Manticore}(\cdot) := \text{Manticore}[M_{(1)}, ..., M_{(K)}](\cdot)$. Let $x = (x_1, ..., x_t) \in \mathcal{V}^t$ be a sequence of $t$ tokens from a set $\mathcal{V}$. The forward pass is computed as follows:

1. Apply the new embedding function $\text{Manticore}_{\text{embed}} : \mathcal{V}^t \to \mathbb{R}^{t \times d_{\max}}$ to the tokens, resulting in a sequence of embeddings denoted $x_{\text{embed}} = \text{Manticore}_{\text{embed}}(x)$.

2. Take forward passes through $L$ Manticore blocks, each with dimension $d_{\max}$, concretely, we compute $x_{\ell+1} := \text{Manticore}_{\text{Block}}^{(\ell)}(x_\ell)$, where $x_1 := x_{\text{embed}}$.

3. Pass $x_{L_M+1}$ into a new task-specific or language modeling head, $\text{Manticore}_{\text{head}} : \mathbb{R}^{t \times d_M} \to \mathbb{T}$, where $\mathbb{T}$ is the appropriate output space for the learning task.

In NAS terms, our search space is over the set of $L \ni \ell$ mixture weights $\alpha^{(\ell)} \in \Delta^{K-1}$. However, **_our search space differs from typical gradient-based NAS techniques_** in the sense that we do not require *discretization* to derive a final architecture after we obtain our mixture weights. Typically, NAS would involve selecting a single sequence of component architecture blocks at each of the Manticore blocks, usually by taking the arg max of the mixture weights. Instead, the mixtures themselves are what characterize Manticore hybrids. Nonetheless, if we were to replace the mixture weights $\alpha^{(\ell)}$ with discrete one-hot vectors, we could derive any of the following: the component model architectures themselves, existing hybrid architectures, and 'frankenmerged' models (Goddard et al., 2024).

## 2.2 How To Use Manticore

With Manticore, we can automatically select language models without training every model in the search space, automatically construct pretrained hybrid architectures without significant trial-and-error, and program pretrained hybrids without full training. In this section, we discuss the details of how Manticore can be used in each of these three usage scenarios.

**Training hybrids from scratch.** *Manticore can be used to automatically select LMs without training all of the LMs in the search space.* Our selection technique is simple: inspired by gradient-based NAS techniques (Liu et al., 2019) and treating the mixture weights as our 'architecture parameters,' we proceed in two steps: 1. train mixture weights along with all other parameters, and 2. freeze the mixture weights and retrain the rest of the parameters from scratch. Unlike NAS, we found that in many pretraining settings, it was sufficient to stop at 1. and forgo retraining. In our pretraining experiments, we use randomly-initialized GPT-Neo (Black et al., 2021) and Mamba (Gu & Dao, 2023) as component models without projectors, and separately experiment with a subset of blocks from MAD (Poli et al., 2024).

**Fine-tuning pretrained hybrids.** *Manticore can be used to create and fine-tune pretrained hybrids.* We create pretrained hybrids as follows: begin with a set of pretrained models, replace their LM heads and embeddings with a single randomly initialized LM head and embedding layer, and pretrain the projectors on a small amount of general language data such as FineWeb (Penedo et al., 2024) while keeping the original component model weights frozen.[3] To fine-tune the pretrained hybrids on downstream task data, we first search for mixture weights by training all of the parameters simultaneously, we freeze the mixture weights, rewind the component models and projectors to their pretrained state, and fine-tune. **This procedure completely sidesteps large-scale pretraining of new hybrids.**[4] In our synthetic experiments, we create pretrained Manticore hybrids from pretrained GPT-Neo-125M (Black et al., 2021) and Mamba-130M (Gu & Dao, 2023) models, while for our experiments on real natural language data, we opt for pretrained Pythia-410M (Biderman et al., 2023) and Mamba-370M (Gu & Dao, 2023) as component models.

**Programming hybrids.** Excitingly, there are cases in which we can program Manticore mixture weights by using external information to predict them. We consider two scenarios.

---

[3]We found that 100M tokens sufficed for projector pretraining.
[4]We include an extensive FLOPs analysis and a discussion of comparable baselines in the Appendix.

If we know that a component model has blocks that are incompatible with the target task—e.g. resulting from sequence length constraints—we can omit these blocks by setting their mixture weights to 0. Otherwise, we can predict good mixture weights by searching on a fixed set of proxy tasks. For this, we use MAD tasks (Poli et al., 2024). The MAD tasks are synthetic unit tests that are predictive of hybrid LM scaling laws, but within our framework, **we find that MAD can also be useful for finding pretrained hybrids.** We use the following procedure for programming mixture weights using the MAD tasks. First, run search on the MAD tasks using a smaller, randomly initialized version of our pretrained hybrid. For each MAD task, our search procedure returns a set of mixture weights—we simply average the resulting mixture weights, freeze them, and fine-tune on downstream task data.

## 2.3 Discussion and Design Considerations

Manticore features several intentional design decisions that we make concrete in this section.

**Where Manticore hybrids excel.** It is known that hybrids excel at compositional tasks like finding a token arbitrarily far in the past and then performing a local copy operation—this for instance, necessitates a tradeoff in SSM (Gu et al., 2022) state size and transformer context. Results like these motivate the study of tools like Manticore. For this reason, we expect that Manticore excels at tasks in which the component models are specialized for certain data sources or aspects of the dataset. As a result, many of our experiments in Section 3 feature heterogeneous data sources.

**Design tradeoffs in Manticore.** Manticore requires taking a forward pass with each of its component models, which increases inference cost over the use of a single component model. **This increased inference cost is an explicit tradeoff for not having to pretrain a hybrid from scratch.** In Appendix E, we motivate this tradeoff by showing that the total FLOPs required to produce a Manticore hybrid is dominated by component model pretraining, and that this can be avoided by reusing existing pretrained models. Due to the simplicity of our projector architecture, we also show that the inference cost of Manticore is dominated by forward passes of its component models. This further motivates our comparison to ensembles in Section 3, due to their similar inference and training FLOPs requirements. Finally, Manticore can be scaled to larger component models without significant overhead, as the inference costs scale linearly in the size of its component models.

**Flexibility of search algorithm.** Our search space works best with NAS algorithms that support continuous-valued mixture weights, such as DARTS (Liu et al., 2019), GAEA (Li et al., 2021), and other gradient-based NAS algorithms. This makes our framework particularly flexible in its support for this broad class of NAS algorithms, while leaving room for specialized algorithms to be developed later. In Appendix B, we include an ablation comparing DARTS to the DASH (Shen et al., 2022) search algorithm, along with various other components of the NAS pipeline. These ablations help characterize the desirable traits of NAS search algorithms for Manticore. For the purposes of our experiments, we mainly rely on DARTS (Liu et al., 2019)—an entirely off-the-shelf NAS algorithm—and leave the development of tailor-made hybrid search algorithms to future work.

## 3 Experimental Results

We provide experimental evidence that validates the following claims about Manticore:

- **C1.** Pretrained hybrids can outperform their component models on fine-tuning tasks,
- **C2.** Trained from scratch, Manticore is competitive with existing hybrids and LMs, and
- **C3.** In certain cases, we can program mixture weights without search on the task data.

### 3.1 Fine-Tuning Pretrained Hybrids

We evaluate **C1**, first on a synthetic task, and then on natural language fine-tuning tasks.

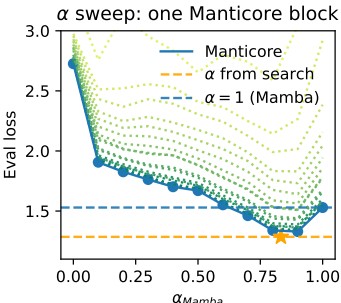 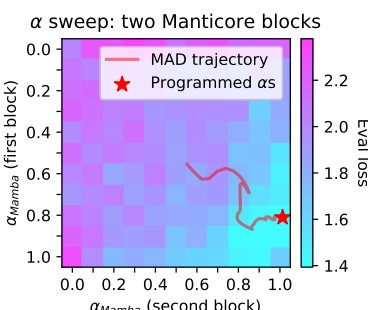

Figure 2: Mixture weight sweeps on Penn Treebank completions using pretrained GPT-Neo-125M and Mamba-130M as our component models. **(Left)** When we create one Manticore block, there is a region of the search space where we improve over Mamba. Here, we denote the loss value and mixture weights found via search using a yellow star and track the loss throughout training in green. **(Right)** The same holds for two Manticore blocks, and our technique for hybrid programming using MAD discovers this region.

**Setup.** We consider a synthetic LM dataset comprising GPT-Neo and Mamba generated completions of text from Penn Treebank (Marcus et al., 1993b). Naturally, we also use pretrained GPT-Neo-125M and Mamba-130M models as component models, creating a single Manticore block with projectors that were pretrained on 100M tokens from FineWeb (Penedo et al., 2024). We search using DARTS, and afterward, we rewind the model weights and projectors to their pretrained states for retraining.

**Results.** Our results are shown in Figure 2 (left). We compare our search results to a sweep over a range of possible mixture weights and find that our search procedure returns the optimal mixture weights, outperforming both Mamba and GPT-Neo. **This confirms our claim that Manticore hybrids can outperform their component models on *synthetic* fine-tuning tasks.** Given that this task comprises two slices that each of our component models should be good at—GPT-Neo should be good at predicting GPT-Neo outputs, and vice versa—we hypothesize that Manticore hybrids are especially well suited to the component models having complementary 'skills' (Chen et al., 2023).

**Setup.** We evaluate on three natural language fine-tuning datasets: Penn Treebank (Marcus et al., 1993b), the Alpaca instructions dataset (Taori et al., 2023), and ELI5 (Fan et al., 2019). We use Pythia-410M and Mamba-370M as our component models, and create a single Manticore block from the blocks of the two models with projectors that were pretrained on 100M tokens from FineWeb (Penedo et al., 2024). As before, we first search for mixture weights, and then we retrain with the fixed mixture weights found by search.

**Results.** Our results are shown in Table 1. Manticore outperforms its component models on Alpaca and ELI5, while it achieves performance between its two component models on Penn Treebank. **This confirms our claim that Manticore can outperform component models on *real* natural language tasks.** The fact that Mamba-370M outperforms Manticore in this setting is not a failure of our framework, as Mamba-370M is included as part of our search space—improving the search procedure beyond off-the-shelf NAS algorithms in order to obtain these high performing models is an interesting direction for future work.

| Task | Pythia-410M (A) | Mamba-370M (B) | Manticore[A, B] |
|---|---|---|---|
| PTB | 0.9099 | **0.8397** | 0.8600 |
| Alpaca | 2.5011 | 2.2999 | **2.1779** |
| ELI5 | 4.1260 | 3.9414 | **3.9331** |

Table 1: Manticore on language tasks using Pythia-410m and Mamba-370m component models. The best test losses are **bolded** and the second-best are underlined.

Figure 3: Mixture weight sweeps using Pythia-410M and Mamba-370M component models. NAS algorithms often locate regions of the search space that outperform component models and a learned ensemble baseline.

**Setup.** Building on the previous setup for natural language tasks, we perform a sweep over the $\alpha$ parameter corresponding to Mamba in our search space, and compare the results of the sweep to off-the-shelf NAS algorithms: DARTS (Liu et al., 2019) (Manticore's search procedure), GAEA (Li et al., 2021), and DASH (Shen et al., 2022). In order to compare Manticore to a method with comparable inference cost, we also consider an ensemble baseline where the ensemble weights are learned during training. For three datasets, 50% of the documents are drawn from the Alpaca (Taori et al., 2023) dataset to artificially induce heterogeneity—we hypothesize that Manticore hybrids are well-suited to such settings—if Manticore's component models specialize in different subsets of a dataset, then Manticore should achieve improved overall performance on the combined dataset.

**Results.** Our results are shown in Figure 3. We find that in all but one setting (NI Chinese QA + Alpaca), at least two of the NAS algorithms that we evaluate recover a model that outperforms its component models. Furthermore, on five of the datasets, at least one NAS algorithm outperforms or matches the best model found during the sweep. Manticore also substantially outperforms the ensemble on all tasks. **This is further evidence for our claim that Manticore outperforms component models on natural language, and demonstrates that NAS algorithms can find performant pretrained hybrids in our search space.**

## 3.2 Training Hybrids from Scratch

For **C2**, we compare to prior hybrids on MAD and non-hybrid models on LRA and MAD.

**Setup.** We compare training Manticore from scratch to training existing hybrid architectures on MAD tasks. We begin with two hybrid architectures from the literature: Mambaformer (Park et al., 2024), which combines Mamba and attention blocks, and the striped multi-head Hyena + Mixture-of-Experts (MoE) MLP architecture that was shown to perform well on the MAD tasks (Poli et al., 2024). We compare these two baselines to a Manticore hybrid combining three component models: striped multi-head Hyena + MoE-MLP, a transformer, and Mamba. We use two blocks for each of these architectures, creating two Manticore blocks. Again, we search for mixture weights and then retrain.

**Results.** The results of this experiment are shown in Table 2 (left). We outperform the striped multi-head Hyena + MoE model from the MAD paper, and we approach the performance of Mambaformer on all but one task. **This validates the claim that Manticore hybrids, trained from scratch, compete with *existing hybrids*.** Despite Mambaformer not being a component model, it is in our search space, and we again speculate that improvements in search would lead to its recovery.

**Setup.** We compare Manticore hybrids to their component models on LRA, when trained from scratch. We use GPT-Neo and Mamba component models of similar sizes to those in Tay et al. (2021) to create Manticore hybrids, while keeping the number of blocks the same between the component models. In these experiments, we create a Manticore block for every block in the component models, ranging from 3 to 6 Manticore blocks.

| Task | Starting from existing hybrids | | | Starting from non-hybrids | | |
|---|---|---|---|---|---|---|
| | SMH Hyena + MoE-MLP (A) | Mamba-former (B) | Manticore | GPT-Neo (C) | Mamba (D) | Manticore [C, D] |
| Ctx. Recall | 3.7153 | **0.0020** | 0.0048 | 4.0771 | 4.1858 | **4.0768** |
| Fuzzy Recall | **4.1714** | **4.1712** | 4.1750 | 4.4384 | 4.8097 | **4.2797** |
| Noisy Recall | 4.1643 | 4.1646 | **4.1607** | 4.1843 | 4.2605 | **4.1823** |
| Sel. Copy | 1.8021 | **0.0005** | 0.0171 | 1.0470 | 3.7765 | **0.9478** |
| Mem. | 8.8353 | **5.2179** | 8.9254 | 4.6110 | 5.2281 | **4.1367** |

Table 2: Results for training from scratch on MAD tasks. **(Left)** Manticore matches the performance of existing hybrids on all but one task. **(Right)** Manticore improves over non-hybrid component models. (Both) best losses are **bolded** and second best are underlined.

**Results.** Our results are shown in Table 3. We outperform component models on all tasks except for IMDb. **This validates the claim that Manticore hybrids, trained from scratch, compete with *existing LMs*.**

| Task | GPT-Neo (A) | Mamba (B) | Manticore[A, B] |
|---|---|---|---|
| ListOps | 37.90 | 20.65 | **38.70** |
| IMDb | 59.62 | **87.74** | 72.44 |
| CIFAR10 | 39.37 | 20.81 | **43.15** |
| Pathfinder32 | 89.41 | 85.76 | **91.45** |
| Pathfinder-X | N/A* | **75.50*** | **75.50*** |

Table 3: Manticore trained from scratch on LRA using GPT-Neo and Mamba component models. Best accuracies are **bolded**. *GPT-Neo does not support the Pathfinder-X sequence length requirement, so its mixture weight is 0 and Manticore reduces to Mamba.

**Setup.** Next, we compare Manticore to non-hybrid architectures trained from scratch on the MAD tasks. For these experiments, our component models use the default architecture and training settings used in MAD. We compare two-block GPT-Neo and Mamba models to a Manticore hybrid using a single Manticore block.

**Results.** Our results are shown in Table 2 (right). Manticore outperforms GPT-Neo and Mamba on all of the MAD tasks in this setting. **This provides further evidence for our claim that Manticore hybrids compete with *existing LMs* when trained from scratch.** It is conceivable that our larger Manticore hybrids simply perform better than component models due to their size—however, we find that post-search discretization and retraining tends to result in similar performance, but reduces the model size by roughly half. We include an ablation of post-search discretization in the Appendix.

### 3.3 Programming Hybrids

We evaluate **C3** with two types of external data: task metadata such as sequence length requirements, and the use of the MAD tasks as a proxy for search on downstream task data.

**Setup.** As in many of our previous experiments, we used the GPT-Neo and Mamba architectures as component models to our Manticore hybrid. However, this time, we set out to train from scratch on the extremely long-range Pathfinder-X task from LRA, which requires sequence length support greater than that of GPT-Neo. Using this external information about the task, we set the mixture weights for GPT-Neo to 0, which in this case, means that Manticore reduces to Mamba.[5]

**Results.** The results of this experiment are shown in the last row of Table 3. **In the simple case of having access to task metadata, this validates the claim that we can program**

---

[5]Mamba on the LRA is open: https://github.com/state-spaces/mamba/issues/282.

**mixture weights to exclude incompatible blocks.** At the time of writing, we are not aware of prior published Mamba results on LRA despite community interest, which would make our evaluation in Table 3 the first such result. Note that we did not thoroughly tune hyperparameters, so we view this result as a preliminary starting point for the community to build off of, rather than a final answer.

**Setup.** Finally, in the case in which we can actually run all of our component models on our learning task, we explore when we can program the mixture weights using the MAD tasks as a proxy for search, which are intended to be predictive of scaling laws on The Pile (Poli et al., 2024; Gao et al., 2020). We set out to fine-tune a pretrained hybrid comprising GPT-Neo-125M and Mamba-130M, which were both pretrained on The Pile, with two Manticore blocks on our Penn Treebank completions synthetic. We train a scaled-down version of this Manticore hybrid with randomly initialized weights and two blocks per component model on the MAD tasks. This yields mixture weights for each of the MAD tasks—we average them across the tasks, and then fine-tune our pretrained hybrid on Penn Treebank completions using the predicted mixture weights.

**Results.** Our results are shown in Figure 2 (right). We superimpose the predicted mixture weights and mean search trajectory from MAD onto the architecture loss landscape computed on Penn Treebank completions. We find that this procedure recovers a hybrid that outperforms the component models (Mamba, lower right; GPT-Neo, upper left) and substantially outperforms the naive frankenmerges in our search space (upper right and lower left) (Goddard et al., 2024). **This is a scenario in which it is possible to program mixture weights using external sources without performing search on the task data.** Intriguingly, search on the MAD tasks appears to follow the architecture gradient on the *different* downstream fine-tuning task, even though the architecture is scaled-down and trained from scratch on MAD. We hypothesize that programming Manticore hybrids becomes more difficult as the fine-tuning distribution is further from the pretraining distribution, and that the architecture loss landscapes become less similar. This evaluation was carried out on our synthetic PTB completions task, so the fine-tuning dataset should be fairly similar to the pretraining distribution. In our evaluation in Table 1, we find that Mamba outperforms the Pythia component model on English natural language tasks that are further from the pretraining distribution than our synthetic (while both models were trained on The Pile (Gao et al., 2020) which is largely in English, we are not training on completions produced by the models themselves). Finally, our evaluations in Figure 3 use non-English text, which is further from the pretraining data distribution, and we observe no discernible pattern between their loss landscapes—programming $\alpha$ parameters in this scenario is likely challenging.

## 4 Conclusions

We present Manticore, a framework that automates the creation of hybrid models from pretrained models by using projectors and convex combinations to align and combine features from multiple different component models, as well as NAS-inspired search procedures. Manticore is efficient and flexible in its usage; hybrids can be trained from scratch, fine-tuned from pretrained component models, and even programmed with external information and/or proxy tasks. In our experiments with several real/synthetic language modeling datasets and existing component/hybrid models, we find that Manticore hybrids match or outperform existing handcrafted hybrid models in these settings. Any requisite fine-tuning and evaluation is performed with a single, large Manticore model rather than designing new hybrids by hand and pretraining them from scratch, which dramatically reduces the computational cost of designing hybrids.

## Acknowledgments

We are grateful for the support of the National Science Foundation (NSF) (CCF2106707), the Defense Advanced Research Projects Agency (DARPA Young Faculty Award), and the Wisconsin Alumni Research Foundation (WARF).

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

# Appendix

## A    Related work

**Language Model Architectures: Transfomers and Beyond.** Transformers are currently the dominant LM architecture. The success of the "vanilla" architecture introduced by Vaswani et. al. (Vaswani et al., 2017) has led to many proposed variations. The quadratic complexity of the base self-attention operation has inspired the search for alternative architectures that offer comparable performance with subquadratic complexity. One line of work builds off *state-space models*, with variations made to enable language modeling (Poli et al., 2023a;b; Gu & Dao, 2023; Arora et al., 2024). Another line of work involves linear-complexity attention by formulating transformers as RNNs and expressing self-attention as a kernel dot-product (Katharopoulos et al., 2020). Other approaches increase the expressivity of this formulation with data-dependent gating (Yang et al., 2024). Our work does not propose a new architecture. Instead, we focus on the idea *that practitioners should be able to take advantage of new architectures in a transparent way*.

**Neural Architecture Search & Mechanistic Search.** Neural architecture search (NAS) techniques are used to automatically search for optimal architectures. These techniques have produced state-of-the-art models in several different architectures and data domains. Much of the challenge in NAS is the complexity of the search procedures; in the most standard form, NAS involves a difficult bilevel optimization over a large search space. Much effort has been aimed at reducing these costs, often via continuous relaxations of the large search spaces, with efficient, end-to-end differentiable search techniques like DARTS (Liu et al., 2019), GAEA (Li et al., 2021), and DASH (Shen et al., 2022).

Using NAS to discover architectures for language modeling—and especially those that may rival Transformers—has thus far been hard. A promising approach is the MAD framework (Poli et al., 2024) , which uses "*mechanistic* tasks" (synthetic tasks organized around simple principles) to search for high-quality subquadratic architectures. While we do not seek to discover *new* architectures, we are inspired by this approach in our effort to search for *hybrid* architectures.

**Hybrid Architectures.** Perhaps unsurprisingly, there is no single dominant architecture among either standards, like Transformers, or emerging subquadratic architectures. While there are some insights that can be converted into heuristics for model selection, generally, to take advantage of new models, practitioners must exhaustively evaluate all of them on each of their tasks. The cost of doing so has inspired the idea of crafting hybrid architectures that mix components from different approaches, with the goal being to obtain best-of-all-worlds behavior.

Unfortunately, the space of hybrid architectures is already large and only grows with each new proposed approach. Manually crafting hybrids is costly; users must either brute-force the enormous search space or alternatively hand-craft a small candidate set of hybrids in the hope that it includes a reasonably performant choice. Our work provides an efficient alternative to this process.

**Model Merging.** A final prospective approach to using multiple models is *merging*. Merging pretrained models (of the same architecture) has shown promising results (Yadav et al., 2023; Yu et al., 2023; Wortsman et al., 2022; Ilharco et al., 2023; Davari & Belilovsky, 2023; Jang et al., 2024), creating powerful large-scale merges such as SOLAR-10.7B (Kim et al., 2023) and Goliath-120B[6] from two fine-tuned Llama2-70B (Touvron et al., 2023) models. The former two were produced using a trial-and-error-based technique called 'frankenmerging,' introduced in MergeKit (Goddard et al., 2024). Frankenmerging involves stitching together different fine-tuned versions of the same model or, hypothetically, different models. This has inspired efforts to merge models of different architectures using large-scale evolutionary search (Akiba et al., 2024). However, such efforts are still embryonic, with substantial computational drawbacks, requiring many training runs. *Manticore, on the other hand, does not require training a large number of models.*

---

[6]https://huggingface.co/alpindale/goliath-120b

## B  Ablations

**Choice of search algorithm.** By default, we use a form of the single-level DARTS (Liu et al., 2019) search algorithm in all of our experiments requiring search. We optionally evaluate whether or not to take *alternating* update, that is, we alternately take gradient steps in the architecture and model parameters—we treat this choice as a task-dependent hyperparameter. However, there are many alternative NAS algorithms that we could have used for search. In our ablation of the choice of search algorithm, we also evaluate DASH (Shen et al., 2022) on our Penn Treebank completions synthetic—the results of which are shown in Table B. In general, we found that using DASH was unable to recover strong architectures in our search space. We postulate that this is because DASH simply aims to solve a different problem, and is not suited to our search space: namely, DASH is used to search for lower-level operations, rather than LM blocks. We also found that alternating DARTS updates was somewhat helpful, compared to simultaneously updating all of the parameters at once—for our experiments, we treated this choice as a hyperparameter.

| Alternating? | DARTS | DASH |
|---:|:---:|:---:|
| Yes | 1.2854 | 2.5899 |
| No | 1.3635 | 2.5968 |

Table 4: Comparison of NAS search methods on our Penn Treebank completions synthetic.

**Whether or not to discretize after search.** We perform an ablation of whether or not to perform discretization on our MAD task experiments in which we compare to existing hybrids. We find that while discretization can sometimes improve performance, the performance differences are often marginal. If final parameter count is a concern, then discretization is beneficial.

| Task | Manticore (non-discretized) | Manticore (discretized) |
|---:|:---:|:---:|
| Context Recall | **0.0068** | 0.0081 |
| Fuzzy Recall | 4.1764 | **4.1729** |
| Noisy Recall | 4.1628 | **4.1614** |
| Selective Copying | 0.0849 | **0.0006** |
| Memorization | 8.9416 | **8.9402** |

Table 5: A comparison of non-discretized vs. discretized Manticore.

**Amount of projector pretraining.** Finally, we ablate over the amount of projector pretraining. We re-ran our $\alpha$ sweep on our PTB completions synthetic with different amounts of projector pretraining, ranging from 0 to 100M tokens sampled from FineWeb (Penedo et al., 2024). The results of this ablation are shown in Figure 4. We found that the optimal value of the $\alpha$ parameter stabilizes around 70M tokens used to pretrain the projectors.

## C  Additional MAD results

In the main text of the paper, we presented results comparing Manticore hybrids trained from scratch to existing hybrids from the literature—namely Mambaformer and the Striped MH Hyena + MOE architecture from MAD. Notably, the Striped MH Hyena + MOE architecture was only the second best architecture presented in the MAD paper. We found that their best architecture, the Striped Hyena Experts + MOE model, performed slightly worse on the harder versions of the MAD tasks that we evaluated. We present these results in Table 6.

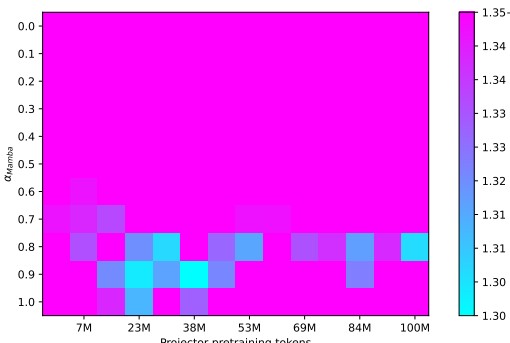

Figure 4: As evaluated on our PTB completions synthetic with Mamba-130M and GPT-Neo-125M, we find that the optimum stabilizes at around 70M tokens of projector pretraining.

| Task | Striped Hyena Experts + MoE-MLP | Striped MH Hyena + MoE-MLP | Mambaformer | Manticore |
|------|---------------------------------|----------------------------|-------------|-----------|
| In-context Recall | 4.0315 | 3.7153 | **0.0020** | 0.0048 |
| Fuzzy In-context Recall | 4.1749 | **4.1714** | **4.1712** | 4.1750 |
| Noisy In-context Recall | 4.1640 | 4.1643 | 4.1646 | **4.1607** |
| Selective Copying | 2.1731 | 1.8021 | **0.0005** | 0.0171 |
| Memorization | 8.8537 | 8.8353 | **5.2179** | 8.9254 |

Table 6: Trained from scratch on MAD tasks, Manticore beats or matches the performance of existing hybrids on all but one task. The best test losses are **bolded** and the second best are underlined.

## D Additional Pathfinder Results

We ran several additional variants of the pathfinder task for which the required sequence length exceeded the maximum supported sequence length of GPT-Neo. We report these results in Table 7.

| Pathfinder task | GPT-Neo (A) | Mamba (B) | Manticore [A, B] |
|-----------------|-------------|-----------|------------------|
| $64 \times 64$, 6 paddles | N/A | 80.40 | 80.40 |
| $64 \times 64$, 9 paddles | N/A | 90.01 | 90.01 |
| $64 \times 64$, 14 paddles | N/A | 86.87 | 86.87 |
| $128 \times 128$, 6 paddles | N/A | 75.50 | 75.50 |

Table 7: Additional Pathfinder results. Note that since these variants of Pathfinder exceed the maximum sequence length of GPT-Neo, we set its mixture weight to be 0 and evaluate using Mamba.

## E On Baselines

The correct set of baselines for Manticore is an interesting and somewhat challenging question. In the main text, we compare to the set of component models used to construct a Manticore hybrid—**in other words, in order for Manticore to be at least as performant as its component models on a task, it must match or beat the performance of the best component model, which implies that** *both* **component models need to be fine-tuned.** This would roughly match the total amount of fine-tuning FLOPs used to train the corresponding Manticore hybrid. However, there are other potential ways to make a comparison; in this section, we will discuss the fairness and availability of baselines corresponding to

different metrics of comparison, and provide a new set of baselines involving ensembles of component models. Specifically, we will address the question of whether the correct comparison is one involving parameter count, training FLOPs, or inference FLOPs.

### E.1 Parameter Count

One proposal is to compare a Manticore hybrid of size $N$ to a pretrained model that is also of size $N$. Manticore combines the weights of existing *pretrained* models to produce a hybrid that is drastically cheaper to generate compared to pretraining a hybrid of the same size from scratch. Off-the-shelf pretrained models of size $N$ are often pretrained up to $D$ tokens corresponding to its Chinchilla optimum (Hoffmann et al., 2022), but information about the amount, mixture, or quality of pretraining data is often unavailable. This makes comparison along the axis of the parameter count alone somewhat challenging—a larger model may well have been trained on more total data than the two smaller component models making up Manticore. In other words, Manticore should not be expected to follow the same pretraining scaling laws as models that were trained from scratch. **Therefore, comparing a Manticore hybrid and a pretrained model of the same size is not necessarily a fair comparison, when considering model size alone. Furthermore, pretrained models of a specific predefined size $N$ are not even guaranteed to exist.**

### E.2 Training FLOPs

Another option is to make a comparison along the axis of total training FLOPs, which would include pretraining FLOPs, fine-tuning FLOPs, and any additional FLOPs incurred when generating a Manticore hybrid. Suppose we create a Manticore hybrid from two component models of sizes $N_1$ and $N_2$, which have been pretrained using $T_1$ and $T_2$ tokens, incurring roughly $6N_1T_1$ and $6N_2T_2$ FLOPs, respectively (Kaplan et al., 2020). With Manticore, we incur FLOPs from two sources: projector pretraining and fine-tuning. In our experiments, we use $T_{\text{proj}} = 100$M tokens of general data for projector pretraining, and saw in Figure 4 that we likely didn't even need this much. Nonetheless, 100M tokens is substantially smaller than the typical amount of pretraining data, so we can assume that $T_{\text{proj}} = 100$M $<< \min\{T_1, T_2\}$, and since the pretrained projectors can be reused, this cost can be amortized over many future fine-tuning runs. Manticore then involves fine-tuning on some small amount of downstream tasks-specific data comprising $T_{\text{ft}} << \min\{T_1, T_2\}$ tokens. So then, the total amount of training FLOPs involved end-to-end in producing a Manticore hybrid is

$$6N_1T_1 + 6N_2T_2 + (6N_1 + 6N_2)T_{\text{proj}} + (6N_1 + 6N_2)T_{\text{ft}} = O(6N_1T_1 + 6N_2T_2),$$

meaning that the total training FLOPs is dominated by the pretraining of the component models. **Our experiments in the main text compare Manticore to the better of the two component models, which means that both component models need to be fine-tuned (i.e., the baseline comprises 'both' component models). Therefore, if the projector pretraining FLOPs are amortized over many fine-tuning runs, Manticore roughly matches the baseline in terms of training FLOPs.** That is, this baseline and Manticore effectively requires $6N_1T_1 + 6N_2T_2 + (6N_1 + 6N_2)T_{\text{ft}}$ FLOPs.

### E.3 Inference FLOPs

It is true that our baselines in the main text (which are pairs of component models) are cheaper in terms of inference FLOPs compared to Manticore. In fact, Manticore effectively doubles the inference FLOPs by requiring forward passes through both component models. Here, we include an analysis of inference FLOPs showing that the contribution of the projectors is negligible, and we present an additional baseline—combining the component models into an ensemble that is fine-tuned simultaneously using the same fine-tuning budget as Manticore.

**Inference FLOPs analysis.** First, we will compute the general form of the inference FLOPs requirement for a component model. Let $d$ be the embedding dimension, let $t$ be the sequence length, let $L$ be the number of blocks, let $v = |\mathcal{V}|$ be the size of the vocabulary set

for our downstream task, and let $B(d, t)$ be the inference FLOPs requirement for the blocks in the component model. Then the inference requirement for a single token prediction from the component model is computed by summing the FLOPs requirements from looking up an embedding, computing forward passes through a sequence of blocks, and generating the final logits. That is, we obtain the following:

$$O(1 + LB(d, t) + dv) = O(LB(d, t) + dv).$$

For a Manticore hybrid, assume that we have $K = 2$ component models, $M_1$ and $M_2$, as well as their projectors. Without loss of generality, assume that the embedding dimensions, $d$, and the number of blocks, $L_M$, in the component models are the same. Let $L << L_M$ be the number of *Manticore* blocks, which is typically constant with respect to the number of blocks in each of the component models $L_M$ (in our experiments, $L$ was set to 1 or 2). Let $B_{M_1}(d, t)$ and $B_{M_2}(d, t)$ be the FLOPs requirements of individual blocks from $M_1$ and $M_2$ respectively, and let $B_{\text{proj}}(d, t) = O(td^2)$ be the FLOPs requirement of projector usage. Note that typically, $B_{\text{proj}}(d, t) = O(td^2) \leq B_{M_*}(d, t)$, as many types of blocks involve a dimension-mixing operation such as an MLP, which has a larger FLOPs requirement than $O(td^2)$, or a sequence mixer that has quadratic or log-linear dependence on $t$, rather than the linear dependence of $B_{\text{proj}}(d)$. Then the FLOPs requirement of each Manticore block is as follows:

$$O\left(\frac{L_M}{L}(B_{M_1}(d, t) + B_{M_2}(d, t)) + 4B_{\text{proj}}(d, t)\right),$$

and along with the token embedding and the logits output, we have

$$O(1) + L * O\left(\frac{L_M}{L}(B_{M_1}(d, t) + B_{M_2}(d, t)) + 4tB_{\text{proj}}(d, t)\right) + O(dv)$$
$$= O\left(L_M B_{M_1}(d, t) + L_M B_{M_2}(d, t) + LB_{\text{proj}}(d, t) + dv\right)$$
$$= O\left(L_M B_{M_1}(d, t) + L_M B_{M_2}(d, t) + td^2 L + dv\right)$$
$$= O\left(L_M B_{M_1}(d, t) + L_M B_{M_2}(d, t) + dv\right),$$

where the final step comes from $L << L_M$ and the assumption that $B_{\text{proj}}(d, t) = O(td^2) \leq B_{M_*}(d, t)$. **This inference cost is the same as inference with both component models. This motivates another baseline: ensembles of component models, which we evaluate next.**

**Comparison to ensembles.** We compare the fine-tuning performance of Manticore to ensembles of component models on the six tasks shown in Figure 3. Starting with pretrained Pythia-410M and Mamba-370M models, we construct our ensemble as follows: for each token prediction, we mix the output probabilities from Pythia-410M and Mamba-370M with equal weighting of 0.5, and then we fine-tune the entire mixture end-to-end on the downstream task. We present the results in Table 8. **The ensemble baseline underperforms Manticore and the best component model on all tasks—we suspect that this could be related to overfitting.**

## F   Hyperparameters

In this section, we discuss our hyperparameters and our experimental setup. Code implementing our experiments can be found at https://anonymous.4open.science/r/manticore-anon.

### F.1   Fine-Tuning Pretrained Hybrids

**Penn Treebank completions synthetic.** For model weights, we use the AdamW (Loshchilov & Hutter, 2019) optimizer with a linear learning rate schedule with an initial learning rate of $5e - 5$. For mixture weights, we use the AdamW (Loshchilov & Hutter, 2019) optimizer with a linear learning rate schedule with an initial learning rate of 0.005 and use alternating updates.

| Task | Pythia-410M (A) | Mamba-370M (B) | Ensemble [A, B] | Manticore [A, B] |
|---|---|---|---|---|
| Es. + Alpaca | 1.819 | 1.704 | 2.172 | **1.664** |
| Ch. + Alpaca | 3.729 | 3.447 | 3.854 | **3.369** |
| Vi. + Alpaca | 2.130 | 2.004 | 2.173 | **1.980** |
| NI non-En. | 1.764 | 1.560 | 1.652 | **1.530** |
| OpenOrcha | 1.570 | 1.576 | 1.756 | **1.553** |
| XQuAD Ar. | 0.205 | 0.207 | 0.533 | **0.201** |

Table 8: Comparison between Manticore, its component models, and an ensemble of its component models on the tasks from Figure 3. For Manticore, we show the best performance achieved across our sweep from Figure 3. Ensembling the component models does not improve performance, but creating a Manticore hybrid does lead to improved performance.

**Fine-tuning on language tasks.** For model weights, we use the AdamW (Loshchilov & Hutter, 2019) optimizer with a linear learning rate schedule with an initial learning rate of $5e - 5$. For mixture weights, we use the AdamW (Loshchilov & Hutter, 2019) optimizer with a linear learning rate schedule with an initial learning rate of 0.005 and use simultaneous updates.

### F.2 Training Hybrids from Scratch

**Comparison to existing hybrids on MAD.**

We provide the hyperparameters and training details for our MAD evaluations from Section 3.2

Existing hybrids were trained with a hyperparameter grid search over the space $[1e - 4, 5e - 4, 1e - 3]$ for learning rate and $[0.0, 0.1]$ for weight decay, similar to the procedure in MAD (Poli et al., 2024).

Manticore is trained in two stages. In the first stage, we train the model and architecture weights in the alternating schedule utilized in DARTS (Liu et al., 2019). In this stage, we perform a hyperparameter grid search of the space $[1e - 4, 5e - 4, 1e - 3]$ for model weight learning rate, $[1e - 4, 1e - 4]$ for architecture weight learning rate, and $[0.1]$ for weight decay. In the second stage, the architecture weights are frozen and we train only the model weights using the best learning rate found in the first stage.

**Evaluation on LRA.** We provide the hyperparameters and training details for our LRA evaluations.

- **ListOps.** We trained all models for 5000 steps. GPT-Neo used 8 attention heads, 6 blocks, an embedding dimension of 512, and a feed-forward network (FFN) dimension of 2048. Mamba used 12 blocks with a model dimension of 512. The vocabulary size was 18.

- **IMDb.** We trained all models for 25 epochs with a batch size of 32. GPT-Neo used 8 attention heads, 6 blocks, an embedding dimension of 512, and an FFN dimension of 2048. Mamba used 12 blocks with a model dimension of 512. The vocabulary size was 129.

- **CIFAR10.** We trained all models for 10 epochs. GPT-Neo used 4 attention heads, 3 blocks, an embedding dimension of 64, and an FFN dimension of 128. Mamba used 6 blocks with a model dimension of 64. The vocabulary size was 256, corresponding to the pixel value range of the grayscale image.

- **Pathfinder32.** We trained all models for 10 epochs. GPT-Neo used 8 attention heads, 4 blocks, an embedding dimension of 128, and an FFN dimension of 128. Mamba used 8 blocks with a model dimension of 128. The vocabulary size was 256, corresponding to the pixel value range of the grayscale image.

**Comparison to non-hybrids on MAD.**

We use two blocks each from GPT-Neo and Mamba, each with a model dimension of 128. We train for 200 epochs and select the best performance during training, as all of the models overfit across the board. We use the AdamW (Loshchilov & Hutter, 2019) optimizer with a linear learning rate schedule with an initial learning rate of $5e - 5$.

### F.3 Programming Hybrids

**Mamba evaluation on long Pathfinder tasks.** Due to our limited computation resources, we did not conduct a hyperparameter sweep for the result we presented. We used Mamba with models of a similar size as Pathfinder32, which has 8 layers, 128 as the hidden dimension size, and 256 as the vocab size. The $64 \times 64$, 6 paddles version is trained by 10 Epoch with default HP. The result for other versions is trained with 200 epochs with default HP in Huggingface trainer.

**MAD tasks as a search proxy.** For model weights, we use the AdamW (Loshchilov & Hutter, 2019) optimizer with a linear learning rate schedule with an initial learning rate of $5e - 5$. For mixture weights, we use the AdamW (Loshchilov & Hutter, 2019) optimizer with a linear learning rate schedule with an initial learning rate of 0.01 and use simultaneous updates. For search on the MAD tasks, we train scaled-down versions of GPT-Neo and Mamba each with four blocks, model dimensions of 128, and no projectors.

### F.4 Pretraining Projectors

For all non-frozen weights (i.e., projectors, mixture weights, embeddings, and the LM head), we use the AdamW (Loshchilov & Hutter, 2019) optimizer with a linear learning rate schedule with an initial learning rate of $5e - 5$.

## G Data and MAD Task Parameters

We provide a more detailed description of the datasets that we use in our experiments. We perform our experiments on a range of synthetic and real tasks that measure various aspects of modern LM capabilities. We discuss the specific datasets that we use in our experiments below. **MAD synthetics.** The MAD synthetic datasets are a set of tasks introduced by Poli et al. (2024) to systematically evaluate the design space of LMs. These tasks are designed to serve as proxy unit tests for rapidly prototyping of new hybrid LM architectures. In our experiments, we use harder variants of the MAD tasks, in which we use a larger vocabulary size of 128 instead of the default 16 for most of the tasks, along with fewer training examples. For simplicity, we omit the compression task as it requires the use of encoder-decoder architectures.

- **In-context recall.** MAD utilizes a multi-query associative recall task, challenging models to retrieve values linked to keys within input sequences, testing their in-context learning ability across randomly shuffled mappings. We use a vocab size of 128 and 800 training examples.

- **Fuzzy in-context recall.** This is a variant of in-context recall to assess a model's ability to semantically group adjacent tokens. Variable-length keys and values are randomly paired, testing the model's capacity for fuzzy recall. We use a vocab size of 128 and 800 training examples.

- **Noisy in-context recall.** This is an adaptation of in-context recall to evaluate a model's capacity to disregard irrelevant information. This involves inserting tokens from a separate vocabulary randomly among key-value pairs, enhancing the memorization challenge. We use a vocab size of 128, a noise vocab size of 16 with 80% noise, and 800 training examples.

- **Selective Copying.** MAD employs a selective copying task to evaluate a model's ability to remember and replicate specific tokens from an input sequence while disregarding randomly inserted noise tokens, emphasizing the preservation of token order. We use a vocab size of 128 with 96 tokens to copy, and 800 training examples.

- **Memorization.** MAD assesses language models' factual knowledge retention through a memorization task, where models learn fixed key-value mappings without in-context computation, testing pure memorization ability. For this task, we use a vocab size of 8192.

**Long Range Arena.** Long Range Arena (LRA) (Tay et al., 2021) is a benchmark consisting of various tasks of different modalities that evaluate how well models can learn long-context data. For simplicity, we omit byte-level document retrieval as it requires two forward passes per example.

- **Long ListOps.** This task is designed to understand whether the architecture is able to model hierarchically structured data in a long-context (Nangia & Bowman, 2018).
- **Byte-level text classification.** This task attempts to test the model's ability to deal with compositionality as in the real world, the model needs to compose characters into words and words into higher-phrases in not so well defined boundaries making it a challenging task, we use IMDB dataset(Maas et al., 2011) in the LRA paper (Tay et al., 2021).
- **Image classification on a sequence of pixels.** This task aims to understand whether a model is able to capture the 2D spatial structure when presented with a flattened 1D version of an image to classify, we use pixel information from CIFAR10(Krizhevsky, 2009) dataset.
- **Pathfinder.** This task helps to understand whether a model can reason about whether the given 2 dots in an image are connected by a path having dashes or not. The sequence length is 1024 i.e a 32x32 image is flattened and provided as input to the model (Linsley et al., 2018; Kim et al., 2020).
- **Pathfinder-X.** An extreme version of Pathfinder with a higher resolution, such as 64x64 and 128*128, which results in a sequence length of up to 16K

**Penn Treebank completions.** We generate a synthetic dataset of generated text from pretrained GPT-Neo-125M (Black et al., 2021) and pretrained Mamba-130M models (Gu & Dao, 2023). We prompt both models using the first four words of every example in the Penn Treebank (Marcus et al., 1993b) validation set, which yields two natural slices of our dataset: sentence completions generated by GPT-Neo and those generated by Mamba.

**Natural language tasks.** We evaluate the ability to fine-tune Manticore on natural language datasets. Specifically, we evaluate on Penn Treebank (Marcus et al., 1993a), the Alpaca instruction tuning dataset (Taori et al., 2023), and an i.i.d. split of the ELI5 training set (Fan et al., 2019). Additionally, we use 100M tokens from the FineWeb dataset (Penedo et al., 2024) to pretrain our projector weights. We describe all other natural language datasets that we use in our evaluations below.

- **NI Spanish QA + Alpaca.** This is from the Natural Instruction dataset v2.8 downloaded from https://github.com/allenai/natural-instructions/releases, we picked task 1610 and mixed it with equal numbers of randomly selected samples from the Alpaca dataset to create a bilingual dataset that contains Spanish Q&A along with English instructions.
- **NI Chinese QA + Alpaca.** This is similar to the previous dataset, except we pick task1570, which is Q&A that input/output language are Chinese.
- **MLQA Vietnamese + Alpaca.** This dataset is a subset of MLQA (MultiLingual Question Answering)(https://huggingface.co/datasets/facebook/mlqa) in which both the inputs and outputs are in Vietnamese, and mixed with equal numbers of randomly selected samples from Alpaca dataset to create a bilingual dataset.
- **OpenOrcha.** We randomly sample 10,000 samples from the OpenOrcha dataset containing Japanese translations from https://huggingface.co/datasets/atsushi3110/cross-lingual-openorcha-830k-en-ja, to form a Japanese Q&A dataset.
- **NI all non-English QA.** There are six Q&A tasks in the Natural Instructions dataset such that both their input and output language is non-English—we combine all of them to form a new dataset containing non-English Q&A.
- **XQuAD Arabic.** The Arabic Q&A part from XQuAD (Cross-lingual Question Answering Dataset), from https://huggingface.co/datasets/google/xquad.

## H A Call for Action & Community Recommendations

Throughout our research process, we noted a handful of opportunities that help to democratize LM research. Should these opportunities be taken up by the research community, we believe they could help to democratize and help to decentralize community-driven LM research, all which enabling further research on pretrained hybrids.

**A search engine for pretrained models.** Surprisingly, we were unable to easily search for pretrained LMs of certain sizes or with certain properties (using Huggingface or otherwise). Tools like this should exist: this would not only significantly democratize LMs, but it would help to reduce monopolies on LM releases and usage, and thereby decentralize LM research.

**Standardized, block-structured LM implementations.** We found that standard tools such as Huggingface and PyTorch were insufficient to cleanly access intermediate activations across several model implementations. This could be resolved by adopting standard implementations or structures for LMs that share the common block structure that we describe in Section 2.1. Instead, our solution was to fork implementations of several Huggingface models, which is time-consuming, error-prone, and non-scalable. A solution to this problem would enable and encourage further research on pretrained hybrid models, which in turn helps to democratize LM research.

**Removing tokenizers from LM pipelines.** We believe that there are too many possible tokenizers, and that tokenizers have a significant potential to introduce merge conflicts in model merging/pretrained hybrid pipelines. In response to this challenge, in our work, we simply chose an arbitrary tokenizer and relearned our embeddings and LM head from scratch in all of our experiments. Possible solutions to this problem would be: as a community, we agree on a standard (small) set of tokenizers, or we eliminate tokenizers altogether by learning character or byte-level LMs.

## I Limitations

At various points in Section 3, we described limitations with using DARTS (the off the shelf NAS search algorithm that we used) for search, in that it was not always able to recover the best architecture in the search space. A potential limitation of Manticore is that it relies on the existence of good gradient-based NAS search algorithms, potentially tailored to our search space. However, we postulate that this is possible, and we leave the task of developing new search techniques to future work.

## J Compute Resources

We ran our experiments on the following GPU hardware:

- 2x Nvidia RTX A6000 GPUs with 48GB GPU memory hosted locally in a nook in the lead author's house and in a friend's basement.

- 2x Nvidia RTX 4090 GPUs with 24GB GPU memory each hosted locally in other friends' basements.

- 2x Nvidia Tesla V100 GPUs with 16GB GPU memory each hosted on AWS (p3.2xlarge instances).

In total, we estimate that our total number of GPU hours across all experiments (those which failed as well as those included in the paper) amounted to roughly 750 GPU hours. We estimate that less than half of these hours accounted for experiments that were not ultimately included in the paper.

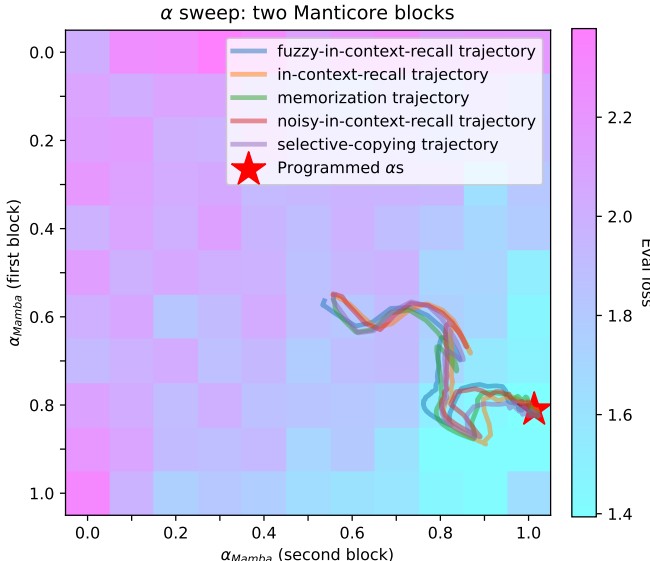

Figure 5: Mixture weight sweeps on Penn Treebank completions using pretrained GPT-Neo-125M and Mamba-130M as our component models. There is a region of the search space where we improve over Mamba when using two Manticore blocks, and our technique for hybrid programming using MAD discovers this region.

## K Expanded Version of Figure 2 (Right)

To show how the architectures evolve over search on all of the MAD tasks in our mixture weights programming experiment, we provide a more detailed version of Figure 2 (Right) – this is shown in Figure 5. Here, we plot the architecture trajectories throughout training on all of the MAD tasks, and superimpose them onto the architecture-loss landscape of the Penn Treebank completions task. The trajectories roughly follow what appears to be a gradient in the loss landscape, and all of the trajectories are roughly similar. We derive our final 'programmed' alphas by taking the average of the final alpha values on each of the MAD tasks, after training.

