# OpenReview forum: "Pretrained Hybrids with MAD Skills"
_colmweb.org/COLM/2025/Conference — COLM 2025_

### Official Review · Reviewer_8jx8 · 2025-05-02

**Rating:** 5
**Confidence:** 4
**Ethics Flag:** 1

**Summary:**

This paper proposes a method for hybridizing pre-trained language models, named Manticore, which introduces input and output projectors as well as a convex combination layer to form a new block (layer) from blocks (consecutive layers) of component blocks.  Formulation is mostly properly explained and experiments show that the proposed method can sometimes lead to the best results in terms of test-set loss.

**Questions To Authors:**

- Why do the authors call the projected vector space (or representations) "a common language"?  Does the resulting vector space suffice the conditions of language, such as discreteness of symbols, structure, recursiveness?

- What is the criterion to choose different sets of component models?  While GPT-Neo-125M, Mamba-130M, Pythia-410M, and Mamba-370M are clearly explained in Section 3.1 and in the second experiment in Section 3.3, the models used in Section 3.2 are not clear.  How large are they?

- What do the green dotted lines and star icon in Figure 2 mean? -> to be removed.

- Could you show the information to support the statement in ll.288-289? -> explained in Appendix.

**Reasons To Accept:**

- The proposed method can be a good alternative to existing methods, such as Mixture-of-Experts (MoE) approach and simple model ensembling, as it can combine component models with different architectures.

**Reasons To Reject:**

- The evaluation results are not tangible with only test-set loss.  I have no idea how good a loss value is in a particular downstream task, except for long-range arena.

- While the authors claim the superiority of the proposed method, the significance of score gains is never tested.

- Selection of the component models in the experiments is not explained, the number of Manticore blocks are limited up to two, the number of component models are also limited to three.  As such, the applicability/generality of the proposed is not fully verified.

- Presentation has some rooms for improvements.  For instance, $L_{d_{M_{(k)}}}$ in l.114 should be $L_{M_{(k)}}$.  All the Tables, presumably except for Table 4, should be transposed so that the numbers to be compared are vertically aligned.  Figure 3 and Table 1 are presented in a wrong order.  Many (at least 12) references miss their source.  Some statements are not understandable.

---

> ### Author Response · Authors · 2025-06-03
>
> Thank you for noting that Manticore is a good alternative to existing methods!
>
> **On downstream task metrics.** We appreciate this feedback. **Table 3 in our draft reports accuracy scores for Long Range Arena (LRA) tasks.** For our language modeling evaluations, we follow standard practice of reporting loss/perplexity as these directly measure model quality for generation tasks. We would be happy to add perplexity values (exp(loss)) alongside loss values if this helps interpretability. Our focus is on comparing Manticore to its component models, which our results demonstrate effectively.
>
> **On the significance of scores.** Indeed, while we do not have multiple seeds for all experiments due to compute constraints, our experimental design uses architecture parameter sweeps (Figures 2, 3) that show **performance surfaces rather than point estimates.** These sweeps demonstrate consistent improvements over component models across different mixture weights, providing evidence for robustness.
>
> **On the selection of component models.** We initially selected the GPT-Neo and Pythia family of models to represent transformers because they are fully open source and provide many different sizes to choose from. We chose Mamba because it is a popular alternative to the transformer family of architectures. However, our framework is not limited to these! We outline in Section 2 that Manticore is general enough to support any language model architecture family that is organized in blocks.
>
> **On the number of component models and blocks.** Our analysis is not limited to two component models and two Manticore blocks. **In fact, in 3.2, we actually used three component models,** but for our other experiments, we chose to use two component models for simplicity and because most hybrids only combine components from two different architectures. **For our LRA experiments, we actually used up to 6 Manticore blocks–we will clarify this in the final version.** For many of our other experiments, we chose to use up to two Manticore blocks for simplicity and because they often performed well. We will include a discussion of these choices in the final version.
>
> **On presentation.** Thank you for pointing out these issues. We have fixed the typo, cosmetic issues, and missing arxiv links. **We are also happy to clarify any specific statements you found unclear;** your feedback is appreciated.
>
> **On calling the shared representations a common language.** Our use of the term is through an analogy. We are not referring to formal languages, and we will clarify.
>
> **On the models used in Section 3.2.** For the experiments on the MAD tasks, we use default architecture settings from the MAD codebase [1], which we train from scratch. We will clarify this in the final version.
>
> **On the green dotted lines in Figure 2.** The green dotted lines represent the loss values throughout training. Since this is not central to our analysis, we will remove these for clarity in the final version. Thank you for pointing this out!
>
> **On support for the statement in lines 288-289.** The following line, 290, notes that this is provided in the Appendix. Specifically, this is shown in Table 5 of Appendix B. There, we show that post-search discretization and retraining yields similar results, but since the Manticore hybrids are discretized (meaning that the convex combinations are rounded to the nearest one-hot vector), the model size and inference cost can be decreased by roughly half.
>
> [1] https://github.com/athms/mad-lab

---

> > ### Comment · Reviewer_8jx8 · 2025-06-06
> >
> > > On downstream task metrics. We appreciate this feedback. Table 3 in our draft reports accuracy scores for Long Range Arena (LRA) tasks. For our language modeling evaluations, we follow standard practice of reporting loss/perplexity as these directly measure model quality for generation tasks. We would be happy to add perplexity values (exp(loss)) alongside loss values if this hel
> > >
> > > On the significance of scores. Indeed, while we do not have multiple seeds for all experiments due to compute constraints, our experimental design uses architecture parameter sweeps (Figures 2, 3) that show performance surfaces rather than point estimates. These sweeps demonstrate consistent improvements over component models across different mixture weights, providing evidence for robustness.
> >
> > While Figure 2 shows a concave over alpha, charts in Figure 3 do not follow it.
> > So, it's great if the authors can show the gain by the proposed method is substantial or marginal.
> >
> > > On the number of component models and blocks. Our analysis is not limited to two component models and two Manticore blocks. In fact, in 3.2, we actually used three component models, but for our other experiments, we chose to use two component models for simplicity and because most hybrids only combine components from two different architectures. For our LRA experiments, we actually used up to 6 Manticore blocks-we will clarify this in the final version. For many of our other experiments, we chose to use up to two Manticore blocks for simplicity and because they often performed well. We will include a discussion of these choices in the final version.
> >
> > Thanks for pointing out my misunderstanding; yes the experiment in Section 3.2 uses three component models.
> >
> > > On calling the shared representations a common language. Our use of the term is through an analogy. We are not referring to formal languages, and we will clarify.
> >
> > Not only formal languages but natural languages (including some animal languages) also have the characteristics that I listed in the review.
> > That analogy is still confusing and could only mislead the readers. I'd await a wise decision of the authors.

---

> > ### Author Response · Authors · 2025-06-06
> >
> > Thank you for the response!
> >
> > **On the nonconvexity of Figure 3.** You're right that multiple seeds would strengthen these results. We will include error bars for Figure 3 in the final version.
> >
> > **On the formal languages analogy.** We see your point, and you’re right, we will remove the language analogy to avoid any confusion.

---

### Official Review · Reviewer_mpR2 · 2025-05-11

**Rating:** 6
**Confidence:** 4
**Ethics Flag:** 1

**Summary:**

This paper introduces Manticore, a novel framework designed to automate the creation of hybrid LMs by combining pretrained models from different architectural families. The framework addresses the challenges of manual hybrid design and the inability to leverage pretrained models by employing projectors to translate features between different architectures and using a convex combination to blend these features. Through a series of experiments on synthetic and real natural language tasks, the authors demonstrate that Manticore hybrids can outperform their component models and existing manually designed hybrids.

**Reasons To Accept:**

- This work models hybrid architecture design as a structural search problem, which is both interesting and meaningful. To my knowledge, current approaches to hybrid model design still heavily rely on manual engineering. The development of automated methods in this area could bring significant advantages.
- As described in line 108, it appears that Manticore’s design does not necessarily require a heavyweight linear transformation. I realize this could potentially reduce the number of additional parameters introduced, as well as the overall training cost.

**Reasons To Reject:**

- I believe that, compared to loss metrics, most readers are more concerned with downstream task performance. Therefore, including a broader set of tasks, particularly involving reasoning or long-sequence generation tasks, in Table 3 would better demonstrate the effectiveness of the searched architectures.
- Including case studies that analyze specific model architectures would significantly enhance the paper. Such analyses could shed light on the characteristics of the hybrid architectures discovered by MAD, addressing questions such as: How do these architectures differ from manually designed counterparts? Are there any consistent design patterns or trends? Furthermore, while I noted in the strengths section that Manticore may not introduce substantial additional computation or parameters, it would be valuable for the paper to offer a more comprehensive analysis of the computational and parametric efficiency of the proposed designs.
- The experiments mainly focus on Mamba-370M and Pythia-410M, with limited consideration of other model families or larger-scale models. This may raise questions about the generality and scalability of the proposed approach.

---

> ### Author Response · Authors · 2025-06-03
>
> Thank you for noting that posing hybrid design as a search problem is interesting and meaningful!
>
> **On metrics and the breadth of tasks.** **Table 3 already reports accuracy rather than loss.** Furthermore, Table 3 specifically **shows accuracies** on Long Rage Arena (LRA) tasks, which span different modalities (ListOps for hierarchical reasoning, IMDb for text classification, CIFAR10 for vision, Pathfinder for long-range dependencies). The MAD tasks (Table 2) specifically test basic reasoning capabilities like associative recall and selective copying. Together, these cover reasoning and long-sequence scenarios.
>
> **On analyzing specific architectures.** Our primary architectural insight is that for the majority of our experiments, Manticore produces architectures that mix its component models, rather than defaulting to the component models themselves. Crucially, there are cases in which MAD can be used as a proxy task to produce architectures that do well on tasks that are similar to the MAD tasks. **Figures 2 and 3 both depict the resulting architectures from our experiments in relation to component models.** In particular, we discuss in lines 317-336 how the architecture derived using MAD relates to both of its component models—GPT-Neo and Mamba—and two naive frankenmerged hybrids comprising these two models. The optimal model in this case, is one that partly still uses at least one of the GPT-Neo blocks, rather than defaulting entirely to Mamba. **This suggests that successful hybrids leverage complementary capabilities rather than simply selecting the better architecture for each block.**
>
> **On analyzing the computational and parametric efficiency of Manticore.** We analyze the parametric efficiency as well as the computational costs associated with pretraining and inference in Appendix E.
>
> **On model families and larger models.** Our experiments demonstrate generality across diverse families: Transformers (GPT-Neo, Pythia), State Space Models (Mamba), and alternative architectures (Striped Hyena). **We test sizes from 125M to 410M parameters, which is a parameter range that is particularly relevant as it represents the scale where many novel architectures (SSMs, linear attention variants) are first validated before scaling up.** Our framework (Section 2.1) supports any architecture following the standard LM recipe. We will clarify in the final version that computational costs scale linearly with component model size, making larger models feasible.

---

> > ### Author Response · Authors · 2025-06-06
> >
> > Thank you for your thoughtful review and positive feedback on our work! We've addressed your specific concerns in our response above—please let us know if there are any remaining questions we could clarify.

---

> > ### Comment · Reviewer_mpR2 · 2025-06-10
> >
> > Thank you for your response. I think this submission can be accepted. I will maintain my rating.

---

### Official Review · Reviewer_NHYJ · 2025-05-14

**Rating:** 6
**Confidence:** 4
**Ethics Flag:** 1

**Summary:**

The manuscript presented Manticore, a framework that automatically builds pre-trained hybrids by mixing blocks from heterogeneous language models through linear projectors and softmax-gated convex combinations. the architecture parameters (mixture weights) are optimized with a differentiable NAS routine (DARTS), after which either (a) the hybrid is trained from scratch, (b) pretrained blocks are reused and only projectors and heads are tuned, or (c) mixture weights are “programmed’’ from proxy tasks without any search. Experiments show that Manticore matches or exceeds handcrafted hybrids like MambaFormer on MAD synthetic tasks, beats its component models on Long range arena and several bilingual fine-tuning datasets, and delivers competitive results while requiring only a single fine-tuned model instead of pre-training multiple candidates

**Questions To Authors:**

Can you quantify end-to-end inference latency for Manticore versus a single component model on LRA and indicate whether knowledge distillation or sparsification could recover the lost efficiency?

How robust is the projector design: do nonlinear or deeper projectors significantly improve performance?

**Reasons To Accept:**

The framework re-uses existing pre-trained models by inserting lightweight gated linear projectors, avoiding costly full-model retraining while ensuring feature compatibility; its NAS-style search over block mixtures is principled yet tractable, enables automatic discovery of performant hybrids that equal or surpass hand-crafted baselines across diverse sequence-length benchmarks; the authors further demonstrate programmed hybrids whose mixture weights are predicted from mechanistic proxy tasks.

**Reasons To Reject:**

Search relies on gradient-based DARTS and can occasionally fails to rediscover optimal hand-engineered hybrids, suggesting sensitivity to NAS hyper-parameters and potential instability; inference cost doubles because all component blocks are evaluated before mixing, and the FLOP analysis indicates that projector overhead is small but total compute is still the sum of the components.

---

> ### Author Response · Authors · 2025-06-03
>
> Thank you for noting the advantages of Manticore, including the reuse of existing pretrained models/avoiding costly pretraining, the tractability of NAS usage, and our experiments with programming pretrained hybrids!
>
> **On the reliance on gradient-based NAS.** It is true that in the NAS literature, where the objective is often to derive a discrete architecture from a learned convex combination of components, DARTS can often fail to produce a performant architecture. **However, our search space is continuous, which makes it particularly well-suited to gradient-based NAS techniques.** While DARTS or other search methods may not always find global optima, Figure 3 shows it consistently finds regions that achieve our goal of outperforming individual component models, and Table 4 shows DARTS outperforms DASH in our setting.
>
> **On inference cost.** This is an explicit tradeoff: doubled inference for avoiding expensive pretraining of new hybrids. All other considerations equal (e.g. total training cost and parameter count), a reasonable baseline would be an ensemble, **and Manticore outperforms ensembles.** We will make this point more clear in the final version. Additionally, NAS is well-suited to problems of trading off considerations like inference cost or model size with performance – exploring NAS algorithms that navigate this tradeoff for Manticore is an interesting avenue that we shall also consider.
>
> **On inference, knowledge distillation, and sparsification.** This is a great point! We measured the inference throughput for the IMDb task in LRA, and obtained the following results:
> - Mamba: 283.508 samples/s
> - GPT-Neo: 87.719 samples/s
> - Manticore: 68.853 samples/s.
>
> This demonstrates that in practice, while the inference cost of Manticore is naturally going to be higher than its component models, the overhead remains quite manageable, especially when combining transformers with subquadratic architectures such as Mamba. Note that we also include an analysis of the inference FLOPs in Appendix E.3. We agree that techniques such as knowledge distillation might be useful for reducing this cost further, and adapting techniques such as MOHAWK [1] to our setting might be a useful next step in this research direction. We perform an ablation over post-search discretization and retraining (which is a form of sparsification; this can be found in Appendix B), and we find that it can help in certain cases. Exploring this direction further to improve the inference cost of Manticore is an intriguing direction for future work.
>
> **On projector architecture.** We did not find that deeper projectors with nonlinearities led to improved performance, and they came at the cost of more expensive projector pretraining. For this reason, we kept the projector design simple, while still including a gating mechanism which would allow for the component architectures to still be part of the search space. We will include a discussion of this in the final version.
>
> [1] https://arxiv.org/abs/2408.10189

---

> > ### Comment · Reviewer_NHYJ · 2025-06-05
> >
> > Thanks authors for the response on inference cost, throughput etc.

---

> > > ### Author Response · Authors · 2025-06-06
> > >
> > > Thank you for your response on inference cost and throughput! If there are any remaining questions we could address to help improve your assessment of our work, we would love to address them!

---

### Official Review · Reviewer_VvYH · 2025-05-16

**Rating:** 6
**Confidence:** 4
**Ethics Flag:** 1

**Summary:**

This paper proposes an approach to build hybrid architectures. Existing approaches suffer from two issues: (i) manual design, where a large search space of hybrid architectures is explored manually, often using unreliable intuition and heuristics, and (ii) the necessity of training hybrid architectures from scratch due to incompatibility between blocks from different architectures, instead of reusing pretrained parameters. The proposed approach, Manticore, automates the design of hybrid architectures by building on ideas from differentiable neural architecture search and introduces projectors to align features between pretrained blocks from different architectures. Furthermore, the paper explores opportunities to program hybrids without full training. The experimental results demonstrate that Manticore hybrids are competitive with or outperform existing manually designed hybrids and individual component models across various synthetic and real-world natural language tasks, including the Long Range Arena (LRA) and Mechanistic Architecture Design (MAD) benchmarks. The paper argues that Manticore dramatically reduces the computational overhead associated with designing and evaluating new LM architectures.

**Questions To Authors:**

See weaknesses

**Reasons To Accept:**

- The paper focuses on the important problem of reducing the difficulty and cost associated with designing and training hybrid architectures. Manticore’s ability to reuse pretrained models is a significant step in developing efficient LMs.
- The idea of bridging the feature spaces of different pretrained architectures via linear projections is novel and effective. Also, training the additional components, including gated residuals and mixture weights, does not require task-specific data, which makes it efficient.
- Manticore offers substantial computational savings by enabling the construction and fine-tuning of hybrids without extensive pretraining. The amortization of projector pretraining costs over multiple fine-tuning runs further enhances its efficiency.
- The experimental results largely support the paper's claims, showing that Manticore hybrids can match or surpass the performance of individual component models and existing handcrafted hybrids on a range of tasks, including long-range dependencies and various natural language understanding benchmarks.

**Reasons To Reject:**

- Manticore achieves a performance between its two component models on Penn Treebank, which raises a question about what kind of tasks Manticore can excel at over its components, and suggests a deeper analysis of why some specific tasks do not benefit from the hybrid approach as much as others.
- While the paper focuses on training efficiency, Manticore requires forward propagation on component models, effectively doubling the inference FLOPs. This could be a practical issue for deployment in latency-sensitive applications.
- Manticore relies on the DARTS algorithm for search; there’s no discussion on: (i) the limitations of DARTS in identifying optimal architecture from Manticore’s search space, or (ii) the desirable traits a search algorithm should possess for Manticore.
- While the authors mention simple projectors were sufficient, they do not discuss their limitations, especially for combining widely different architectures, or the tradeoffs with more complex projection architectures.

---

> ### Author Response · Authors · 2025-06-03
>
> Thank you for praising the importance of our problem, the significance of the steps that Manticore takes toward developing efficient LMs, the novelty of our approach, efficiency, and of our results supporting our claims! Your other points are aligned with clarifications that we are already in the progress of integrating into the manuscript, and we respond to them below.
>
> **On the types of tasks that Manticore helps with.** This is a great point that we would love to make more explicit in the final version! Several of our experiments in Figure 3 were on datasets that combine data from multiple sources (e.g. NI Spanish QA and Alpaca). Our empirical results reflect some of the known early set of theoretical findings on hybrid models. Specifically, it is known that hybrids excel at compositional tasks like finding a token arbitrarily far in the past and then performing a local copy operation – this for instance, necessitates a tradeoff in SSM state size and transformer context. Results like these motivate the study of tools like Manticore. For this reason, we suspect that Manticore excels at tasks in which the component models are specialized for certain data sources or aspects of the dataset, which we briefly mention on line 251. We will discuss this further in the final version.
>
> **On the cost of inference FLOPs.** We discuss our FLOPs analysis and additional experimental results related to this in Appendix E. All other considerations equal (e.g. total training cost and parameter count), a reasonable baseline would be an ensemble, **and Manticore outperforms ensembles.** Although latency-sensitive applications are not the focus of our work, we note that for such applications where the inference cost of Manticore (or ensembles) is prohibitive, it might still be beneficial to train a hybrid from scratch.
>
> **On DARTS as the default search algorithm.** We include experimental results that **compare DARTS to other NAS algorithms** – this is shown in Figure 3. We found that between DARTS, DASH, and GAEA, there was no clear winner across all of the tasks, but that for all but one task, at least two of them were effective. In short, we agree with your point (i) that we should discuss the limitations of DARTS for Manticore, but we note that **our framework is flexible and supports other NAS algorithms, while leaving room for specialized algorithms to be developed for pretrained hybrid search spaces. We will highlight this in the final version.** For (ii), we also include an ablation comparing DARTS to the DASH search algorithm, both with and without alternating architecture parameter/shared weights updates in Appendix B, along with whether or not to perform post-search discretization and retraining. **These ablations help characterize the desirable traits of NAS search algorithms for Manticore.**
>
> **On the choice of projector architecture.** This is a great point! We wanted to keep the projector design ***as simple as possible*** to reduce the cost of projector pretraining, while including a gating mechanism that would allow for the component architectures to still be part of the search space. Conceivably, one could use multi-layer projectors with nonlinearities, but we found that this introduced higher cost and complexity without necessarily improving performance. We will include a discussion of this in the final version.

---

> > ### Author Response · Authors · 2025-06-06
> >
> > Thank you for your thoughtful review and positive feedback on our work! We've addressed your specific concerns in our response above—please let us know if there are any remaining questions we could clarify.

---

> > > ### Comment · Reviewer_VvYH · 2025-06-08
> > >
> > > Thank you for your response. The revision will benefit from planned discussions about tasks suitable for Manticore, desirable traits for a search algorithm, and tradeoffs of different projectors.

---

> > > > ### Author Response · Authors · 2025-06-09
> > > >
> > > > We agree, and we are currently incorporating these into the draft! If there are any remaining questions we could address in the meantime to help improve your score, we would love to address them!

---

### Author Response · Authors · 2025-06-03

We thank all reviewers for their thoughtful feedback and constructive suggestions. We are encouraged by the positive reception of our work:

**Reviewer VvYH** praised the importance of our problem, the significance of Manticore's steps toward developing efficient LMs, the novelty of our approach, and our efficiency gains and experimental validation.

**Reviewer NHYJ** highlighted Manticore's advantages in reusing existing pretrained models while avoiding costly pretraining, the principled tractability of our NAS usage, and our experiments with programming pretrained hybrids.

**Reviewer mpR2** noted that modeling hybrid architecture design as a structural search problem is both interesting and meaningful, and appreciated our lightweight design that reduces additional parameters and training costs.

**Reviewer 8jx8** recognized that Manticore provides a good alternative to existing methods like model ensembling, particularly for combining component models with different architectures.

We address each reviewer's specific questions and concerns in our individual responses below—we have already been incorporating the suggested clarifications and improvements into our final manuscript.

---

### Decision · Program_Chairs · 2025-07-08

**Decision:**

Accept

**Comment:**

The paper introduces Manticore, a framework for automatically constructing hybrid language models by combining blocks from pretrained models of different architectures. It uses lightweight linear projectors to align feature spaces and applies a differentiable architecture search (DARTS) to optimize softmax-gated mixtures of blocks. Manticore supports three modes: training from scratch, fine-tuning only projectors and heads, or programming mixtures from proxy tasks without search. Experiments on synthetic and real-world NLP tasks show that Manticore hybrids often outperform their components and handcrafted hybrids, while offering significant training efficiency and architectural flexibility.

Pros:
- The approach automates hybrid LM design, reducing reliance on manual heuristics.
- The approach reuses pretrained blocks efficiently via lightweight linear projectors.
- The approach supports multiple modes: training from scratch, partial fine-tuning, or zero-shot programming.
- It achieves competitive or superior performance to component models and handcrafted hybrids on several benchmarks.
- The approach substantially reduces training cost by avoiding full model retraining.
- The approach enables flexible mixing of heterogeneous architectures through a principled NAS approach (DARTS).

Cons:
- Inference cost is high due to simultaneous evaluation of all component blocks.
- Limited diversity in model families and scale raises questions about generality.
- Evaluation relies heavily on loss metrics, with limited downstream task coverage or statistical testing.
- Paper lacks detailed analysis of discovered architectures and their computational efficiency.

I believe the authors have engaged well with the reviewers and will clarify about the work and its limitations in the final version of the paper.